# Sugar Kelp (*Saccharina latissima*) Seaweed Added to a Growing-Finishing Lamb Diet Has a Positive Effect on Quality Traits and on Mineral Content of Meat

**DOI:** 10.3390/foods12112131

**Published:** 2023-05-25

**Authors:** Vladana Grabež, Hanne Devle, Alemayehu Kidane, Liv Torunn Mydland, Margareth Øverland, Silje Ottestad, Per Berg, Karoline Kåsin, Lene Ruud, Victoria Karlsen, Valentina Živanović, Bjørg Egelandsdal

**Affiliations:** 1Faculty of Chemistry, Biotechnology and Food Science, Norwegian University of Life Sciences, P.O. Box 5003, 1432 Ås, Norway; hanne.devle@nmbu.no (H.D.); karoline.kasin@stami.no (K.K.); lene.ruud@nmbu.no (L.R.); victoria.karlsen02@gmail.com (V.K.); bjorg.egelandsdal@nmbu.no (B.E.); 2Faculty of Bioscience, Norwegian University of Life Sciences, P.O. Box 5003, 1432 Ås, Norway; alemayehu.sagaye@nmbu.no (A.K.); liv.mydland@nmbu.no (L.T.M.); margareth.overland@nmbu.no (M.Ø.); 3Nortura SA, P.O. Box 360, 0513 Oslo, Norway; silje.ottestad@hotmail.com (S.O.); per.berg@nortura.no (P.B.); 4Faculty of Environmental Sciences and Natural Resource Management, Norwegian University of Life Sciences, P.O. Box 5003, 1432 Ås, Norway; valentina.zivanovic@nmbu.no

**Keywords:** growing-finishing lamb diet, seaweed, meat quality traits, minerals

## Abstract

Supplementing ruminants’ diet with seaweed has shown positive effect on meat quality and micronutrients important for human health. The objective of the present study was to investigate the use of *Saccharina latissima* in a lamb diet to improve the eating quality and nutritional value of meat. Six-month-old female Norwegian White lambs (*n* = 24) were fed, 35 days pre-slaughter, three different diets: a control (CON) and two seaweed diets (SW); supplemented with either 2.5% (SW1) or 5% (SW2). The quality properties of longissimus thoracis et lumborum (LTL) and semimembranosus with adductor (SM+ADD) muscles were examined. The dietary inclusion of seaweed reduced cooking loss and shear force of lamb meat, although the effect was not significant at both supplementation levels. SW1 fed lambs showed a significantly (*p* < 0.05) improved meat color stability and antioxidant potential. Seaweed also reduced lipid oxidation (TBARS) and the warm-over flavor in SM+ADD compared to the CON lamb. Seaweed fed lambs showed an increased content of selenium and iodine in LTL, thereby fulfilling the requirements for the label “source of nutrient” and “significant source of nutrient”, respectively. An increased arsenic content in LTL was, however, also observed with seaweed inclusion (to 1.54 and 3.09 μg/100 g in SW1 and SW2 group, respectively). While relevant positive effects were found in meat using seaweed in lamb feed, some optimization of this feed approach will be desirable.

## 1. Introduction

Red meat represents a rich source of high-quality protein and a wide range of micronutrients such as iron (Fe), selenium (Se), zinc (Zn), vitamins A, K, B6, B12, and folic acid, which are required for human health [1]. In recent years, meat production has garnered negative publicity as the consumption of red meat has been associated with increased disease risks [2]. Therefore, the increased demand for meat with improved nutritional value and sustainable production requires new strategies. The nutrient content in meat depends on several factors, of which feed composition is very important. The inclusion of specific nutrients in the finishing cattle diet has been suggested as a cutting-edge strategy to improve the nutrient composition of beef meat [1].

Norwegian lamb meat production relies on the use of open pastures in high-mountain and coastal areas that are unsuitable for plant proteins for human consumption. Although in Norway, most lambs (4–6 months old) are slaughtered in the autumn directly from the mountain pastures, large differences in live weight may sometimes require the fattening of animals with concentrate or the use of cultivated pasture in the growing-finishing phase [3].

In general, seaweed presents a valuable source of bioactive compounds such as the polysaccharides laminarin and fucoidan, polyphenols, and vitamins C, E, and B, and minerals such as Fe, Zn, Se, manganese (Mn), copper (Cu), and iodine (I) [4]. The inclusion of seaweed in lamb growing-finishing diets potentially could enhance the content of specific minerals in meat. Recent studies revealed I deficiency among young and pregnant women in Norway [5]. In addition, there is an indication of Se deficiency in the Norwegian population. Dierick et al. [6] proposed I enrichment of pork meat by seaweed supplementation as a strategy to improve the I supply of humans in Belgium.

Morais et al. [4] reviewed the status of seaweed as ruminant feed and referred positively to the high relevance and positive contribution of many minerals in seaweed to animal health. However, bioaccumulation of toxic elements from seaweed, i.e., arsenic (As) and cadmium (Cd), in the muscle tissue of animals require special attention. In addition, there are very few studies about the effects of seaweed supplementation in lamb diets on fresh and processed meat quality [7,8,9]. Therefore, the present study evaluated a feeding strategy, using increasing levels of the brown seaweed *Saccharina latissima* in growing-finishing lamb diets, in order to enhance the quality traits of fresh meat. The concentrations of desirable minerals and toxic elements in the meat were analyzed. The potential impact of seaweed-supplemented lamb meat on the consumers’ mineral intake will be discussed.

## 2. Materials and Methods

### 2.1. Management and Dietary Treatments

Twenty-four weaned Norwegian White female lambs weighing 37.3 ± 1.6 kg were randomly assigned to three groups (*n* = 8 per group). The three dietary treatments consisted of a control diet (CON; total mixed ration of grass silage mixed with compound feed, rolled barley, and mineral premix) and SW1 and SW2, in which the control diet was supplemented with either 2.5% or 5.0% brown seaweed (*Saccharina latissima*) on a dry matter (DM) basis, respectively. The ingredients and chemical composition of the three diets are presented in Table 1.

The seaweed was chopped to a uniform particle length, wilted by spreading over a plastic sheet in the open air, and frequently stirred to aerate and prevent mold formation. The wilted material was immediately frozen until the final preparation of the experimental diets. The lambs were housed in individual pens (76 cm × 157 cm) with plastic slatted floors and individually fed the respective experimental diets in two meals per day. The amount of refused feed was weighed every day. All lambs had free access to clean drinking water. The trial lasted for 35 days, until the slaughter of the lambs at a commercial slaughterhouse. All animal procedures for the indoor lambs were approved by the committee overseeing the rules and regulations governing animal experiments in Norway under the surveillance of the Norwegian Food Safety Authority (FOTS-ID: 16406). The lamb growth rate was monitored by weekly weighing using a digital scale (BioControl Stationary Reader, SR3000, BioControl AS, Rakkestad, Norway).

### 2.2. Feed Composition

Feed samples were collected on a weekly basis, dried at 59 °C for 48 h, and grinded in a cutting mill (Retsch SM200, Retsch GmbH, Germany) with a 1.0 mm sieve. The samples were analyzed in duplicates. The chemical analyses were performed at LabTek, Department of Animal and Aquacultural Science, NMBU, Ås, Norway. Briefly, gross energy was determined using a PARR 6400 Automatic Isoperibol Calorimeter (Parr Instruments, Moline, IL, USA) [10]. DM, ash, and total nitrogen (N) were determined according to EC Regulation No 152/2009 [11], and crude protein (CP) was calculated as N × 6.25. Neutral detergent fiber (NDF) and acid detergent fiber (ADF) were determined using the filter bag technique in an Ankom 200 Fiber Analyzer (Ankom Technology, Macedon, NY, USA) according to the manufacturer’s instructions. Starch content was determined according to the AOAC method 996.11 [12] using an RX Daytona + spectrophotometer for glucose analysis (Randox Laboratories Ltd., Crumlin, UK).

### 2.3. Slaughtering and Muscle Sampling

All animals were slaughtered on the same day in a commercial slaughterhouse (Rudshøgda, Nortura, SA, Norway). The lambs were electrically stunned (3.2 s at 1.3 A) and electrically stimulated (at 90–100 V AC for a minimum of 60 s) after exsanguination. Hot carcasses were weighed and graded according to the European classification system (EUROP). The carcass weight obtained from the slaughterhouse was used to calculate the dressing percentage ([carcass weight/live weight] × 100). Carcasses were hung separately on hooks and chilled at 2–4 °C for 24 h. Cold carcasses were transported the same day from the slaughterhouse to the Norwegian Meat and Poultry Research Center.

pH values were measured both in longissimus thoracis et lumborum (LTL) and semimembranosus (SM) on the day of carcass dissection (Day 0); the CON and SW2 carcasses were dissected 48 h post-slaughter, and SW1 carcasses 72 h post-mortem due to restricted cutting capacity. The pH values of the SM may have a bias (0.04 units higher) as pH meters had to be changed. However, only minor differences in pH occur between 48 h and 72 h post-mortem [13]. The pH meter (Knicks Portamess 913, Berlin, Germany) equipped with a Hamilton double pore glass electrode was calibrated with buffer solutions (pH 4.0 and 7.0) prior to measurements.

On the carcass dissection day (Day 0), vacuum-packed LTL and SM with adductor (ADD) were transported 40 min to the laboratory in Styrofoam boxes filled with ice. Upon arrival, three slices (2 cm thick) of LTL and SM from the same anatomical region were cut parallel to the fiber direction and stored at 4 °C: two were vacuum-packed for shear force (SF) measurements, and one was slice-wrapped in polyvinyl chloride (PVC) foil for color assessment. The remaining LTL and SM+ADD meat was ground, divided into smaller portions, and vacuum packed. One portion of homogenized LTL and one of SM+ADD was stored at 4 °C for 4 weeks and afterward at –80 °C until thiobarbituric acid reactive substances (TBARS) and 2,2,1-diphenyl-1-picrylhydrazyl (DPPH) analysis. The remaining homogenized LTL and SM+ADD were immediately frozen and kept at −80 °C (Day 0) until analyzed for DPPH, TBARS, warmed-over flavor (WOF), fatty acid methyl esters (FAME), Vitamin B12, minerals, and toxic elements.

### 2.4. Cooking Loss, Shear Force, and Color

Vacuum-packed LTL and SM samples (slice thickness ca 2 cm) on Day 7 (nine days post-mortem) were placed in a preheated water bath at 80 °C and cooked until reaching an internal temperature of 72 °C. After cooking, the samples were cooled in ice-cold water to 20 °C. The next day, cooked meat was removed from the plastic bag, dried with a paper towel, and weight (*W*1). Liquid loss of cooked meat was weight (*W*2). Cooking loss was calculated with the following calculation:(1)Cooking loss %=W2W1+W2×100

Warner–Bratzler shear force (SF) analyses were performed using a Texture Analyzer (Model: TA-HDi, Producer: Stable Micro Systems, Godalming, UK) equipped with a shear cell HDP/BSK and fitted with a 25 kg load cell. Shear measurements were done perpendicular to the muscle fiber direction. The samples were equilibrated to room temperature and 6–10 cylindrical samples (1 × 1 × 4 cm) were cut along the fiber direction, avoiding fat and connective tissue.

Slices of LTL and SM (2 cm thick) cut at Day 0 were allowed to bloom for 50 min at 15 °C, and the color of the surface was then measured. The slice was wrapped in oxygen-permeable PVC film and stored at 4 °C and used for color measurements on Days 3 and 6. CIELAB color coordinates L* (lightness), a* (redness), and b* (yellowness), were measured using a Konica Minolta Spectrophotometer CM 700d (Illuminant D65, 8 mm diameter measurement area, 8-degree viewing angle, Konica Minolta Sensing Inc., Osaka, Japan) with a closed cone. The colorimeter was calibrated using a white ceramic calibration cap (CM-A177). Measurements were made in two places on the surface of the meat covered with PVC foil at two different zones of each sample. Color values were analyzed using Spectra Magic Software (Minolta Inc., Tokyo, Japan). To determine the myoglobin (Mb) states, samples were scanned for absorbance from 400 to 700 nm and the predicted Mb states were calculated [14].

### 2.5. Oxidative Stability

The antioxidant activity of LTL and SM+ADD stored at –80 °C on Day 0 and after 4 weeks of chilled storage was measured using the modified DPPH^●^ method described by Brand-Williams et al. [15]. Briefly, 0.5 g of meat was mixed with 4 mL of ethanolic DPPH^●^ solution (0.05 mg/mL) and incubated for 50 min in the dark, at room temperature. The sample was vortexed, and a 2 mL aliquot was transferred to an Eppendorf tube and centrifuged at 16,464× *g* for 5 min at 20 °C. A sample of 200 μL was transferred to a 96-well plate, and the absorbance was measured at 515 nm at 20 °C using a Synergy H4 Hybrid Multi-Mode Microplate Reader (BioTek Instruments Inc., Winooski, VT, USA).

Lipid oxidation analyses were performed on LTL and SM+ADD samples stored at –80 °C on Day 0 and after 4 weeks of chilled storage. In addition, lipid oxidation was analyzed in meat heated at 72 °C followed by chilled storage for 24 h to accelerate oxidation (warmed-over flavor, WOF). The homogenized sample (2 g) was transferred into a 50 mL Falcon tube, and a volume of 10 mL of stock solution (0.38% thiobarbituric acid (TBA) and 15% trichloroacetic acid (TCA) in 0.25 N HCl) was added. The sample was heated in a water bath at 99.9 °C for 10 min and then promptly cooled in an ice bath to room temperature. An aliquot of 1.5 mL was transferred to an Eppendorf tube and centrifuged at 21,500× *g* for 25 min at 4 °C. A 200 μL aliquot was transferred to a 96-well plate, and the absorbance at 532 nm was measured using a BioTek Synergy H4 plate reader. All samples were analyzed in duplicate and compared to a blank. The 2-thiobarbituric reactive substances (TBARS) are expressed as mg of malonaldehyde (MDA)/kg of meat.

### 2.6. Fatty Acid and Nutrient Analyses

Frozen meat samples (~20 g) were powdered using a laboratory homogenizer (IKA 11 basic Analytical mill, Staufen, Germany). LTL and SM+ADD samples and freeze-dried feed samples of 0.25 g were used for the extraction of fatty acid methyl esters (FAMEs) with hexane [16]. In short, the powdered sample was mixed with 1 mL of C13:0 internal standard (0.5 mg C13:0/mL methanol). The sample was dissolved and hydrolyzed with 0.56 mL of 10 N KOH in water and 4.2 mL of methanol and incubated in a water bath at 55 °C for 1 h including shaking it for 5 s every 20 min. 0.46 mL of 24 N sulfuric acid was added to the cooled sample, and the incubation was repeated. When the sample was cooled, 3 mL of hexane was added, and the sample was mixed for 5 min. The sample was centrifuged to separate the upper hexane layer containing FAMEs, which was transferred into a glass gas chromatography (GC) vial and kept at −20 °C prior to analysis. The FAMEs were analyzed using GC with a flame ionization detector (FID) (Thermo Scientific™, Dreieich, Germany). Chromeleon software (version 7.2.10, Thermo Fisher Scientific, Waltham, MA, USA) was used for instrument control and data processing. For GC a Thermo Scientific™ Trace™ 1310 instrument equipped with an Agilent J&W column (Folsom, CA, USA) suitable for FAME analysis (CP-Sil 88, 100 m, 0.20 µm film thickness and ID of 0.25 mm) was used. The stationary phase was highly polar, containing 88% biscyanopropyl polysiloxane (Agilent J&W, Folsom, CA, USA). Helium (99.99990% from Yara, Rjukan, Norway) was used as the mobile phase at a constant flow of 2.0 mL/min. The temperature program started at 45 °C, and this temperature was held for 4 min, increased to 165 °C (13 °C/min) and held for 45 min, increased to 200 °C (8 °C/min) and held for 15 min, and increased to 215 °C (3 °C/min) and held for 15 min. The temperature was then increased in a final step to 240 °C (30 °C/min). The GC was equipped with a Thermo AI 1310 liquid autosampler (Thermo Scientific™, Waltham, MA, USA). One microliter of the sample was injected into a split/split less injector operated in split mode at a split ratio of 1:10 and an injector temperature of 250 °C. The FAMEs were identified by comparing chromatographic retention times with the retention times of a purchased FAME standard mixture (Mixture ME 100, Larodan, Solna, Sweden) and individual FAME standards. The following individual FAME standards were supplied by Larodan (Solna, Sweden): methyl 13-methyltetradecanoate, methyl 12-methyltetradecanoate, methyl 14-methylpentadecanoate, methyl 15-methylhexadecanoate, methyl 16-methylheptadecanoate, methyl 7(Z)-hexadecenoate, methyl 11(E)-octadecenoate, methyl 13(Z)-octadecenoate, methyl 9(Z),11(E)-octadecadienoate, methyl 7(Z),10(Z),13(Z),16(Z)-docosatetraenoate, methyl 7(Z),10(Z),13(Z),16(Z),19(Z)-docosapentaenoate. Methyl 10(E)-octadecenoate was purchased from Toronto Research Chemicals (Toronto, Canada). The concentrations of FA in growing-finishing diets (g/kg DM) and lamb meat (mg/100 g meat) were determined using a five-point standard curve of C13:0 fatty acid methyl ester.

Vitamin B12 analysis was performed using a VitaFast^®^ microbiological assay with AOAC licensed number 101002 [17]. LTL (1 g) was weighed into a 50 mL Falcon tube and mixed with 20 mL of acetate buffer (pH 4.5), 250 μL of 1% KCN, and 300 mg of taka-diastase (α-amylase from Aspergillus oryzae). The mixture was incubated at 37 °C (Heraeus Incubator, Thermo Scientific, Langenselbold, Germany) for 1 h in the dark with handshaking for 5 s every 20 min. The Falcon tubes were filled up to exactly 40 mL with deionized water, shaken vigorously, incubated in a water bath at 95 °C for 30 min with handshaking for 5 s every 6 min, and then cooled to below 20 °C. An aliquot of 1.5 mL was transferred to a clean Eppendorf tube and centrifuged at 9000× g for 10 min at 20 °C. Then, 0.1 mL of extract was diluted with sterile water and vortexed for 60 s, and 150 μL was pipetted into a 96-well microtiter plate that contained Lactobacillus delbrueckii subsp. lactis and 150 μL of assay medium. The plate was incubated at 37 °C for 48 h in the dark. The absorbance was measured over the range 610–630 nm using a Synergy H4 Hybrid Multi-Mode Microplate Reader (BioTek Instruments Inc.). A cyanocobalamin and a pork liver BCR-487 standard curve were used for B12 quantification. The coefficient of variation was below 10%.

For microelement analysis (Fe, Se, Cu, As, Cd) freeze-dried feed and LTL samples (0.1–0.15 g) were digested with 5 mL of ultra-pure nitric acid and 2 mL of Milli-Q water (Merck Millipore, Burlington, MA, USA) in a Teflon tube (final concentration 10% [*v*/*v*] ultrapure nitric acid after dilution). The digestion was performed at 260 °C for 60 min in an UltraClave (Milestone, Sorisole, Italy). The digested sample was diluted to 50 mL with Milli-Q water. The sample was analyzed on an Agilent 8800 ICP-MS (Agilent Technology). The same digestion procedure was applied to samples, blanks, and reference material (SRM 1577 c [NIST] and CRM DORM 3, NRCC) to confirm the amount of analyte. The quantification of microelement content was carried out with microelements standard curve having a correlation coefficient >0.95.

The iodine in feed samples was extracted using tetramethylammonium hydroxide (TMAH) at 90 °C for 3 h. The sample was centrifuged and diluted to 50 mL with deionized water. The extract was analyzed using an Agilent 8800 ICP-MS (Agilent Technologies, Hachioji, Japan). The quantification of microelement content was carried out with an I standard curve having a correlation coefficient >0.95.

For I analysis in meat, 1 g of LTL was weighed in a 50 mL Falcon tube, and extraction was performed using a graphite block system (DigiPREP, SCP Science, Courtaboeuf, France) with tetramethylammonium hydroxide (TMAH) for 3 h at 90 °C. The sample was centrifuged and filtered through a 0.45 μm filter. Iodine was determined using inductively coupled plasma–mass spectrometry (ICP–MS, Thermo X series II) on a system equipped with a Cetac ASX-520 autosampler. The quantification of the I content was carried out with an I standard curve having a correlation coefficient ≥0.9995.

### 2.7. Statistics

The effect of dietary treatment on average daily weight gain (ADWG), carcass traits, nutrient content, and iodine content in plasma samples was assessed using a one-way analysis of variance (ANOVA). To investigate the effect of cold carcass weight on conformation score, the fit regression model y = β0 + β1 × 1 (×1 cold carcass weight as predictor variables) was applied.

A two-way analysis of variance (general linear model) was used to assess the effect of diet, muscle type, and the interaction between these two as a fixed factors on quality properties (pH, cooking losses, SF, color, TBARS, DPPH, and WOF) and fatty acid classes. Differences between the means were determined using Tukey test (*p* < 0.05). The statistical analyses were performed using Minitab, version 18 (Minitab Inc., State College, PA, USA).

Calculations of Mb states involved predictions from cross-validated partial least square regression models using scatter-corrected data obtained on pure Mb states (Unscrambler X 10.5, Trondheim, Norway) [14].

## 3. Results

### 3.1. Weight Gain and Carcass Traits

Data on average daily weight gain ADWG, carcass and meat quality traits are provided in Table 2. Seaweed supplementation had no effect (*p* > 0.1) on the ADWG, cold carcass weight, or dressing percentage of the lambs relative to the control. In addition, the fatness degree and the carcass conformation were unaffected (*p* > 0.05) by a growing-finishing diet. The conformation of all carcasses ranged between scores of R and U, with the SW2 diet resulting in a greater proportion of carcasses classified as U (50%), whereas the other treatments had a greater proportion of R scores (62.5% and 83.3% for CON and SW1, respectively). Cold carcass weight was positively correlated (*p* < 0.05) with a conformation score. SW1 animals had numerically lower fatness scores due to one outlier carcass. In general, all dietary groups were classified within a medium fatness degree.

### 3.2. Meat Quality

As shown in Table 3, the growing-finishing diet had an effect (*p* < 0.001) on the ultimate pH of meat, with SW1 showing a higher pH than CON and SW2. However, incremental seaweed addition showed a non-linear pH response. Cooking losses were significantly (*p* ≤ 0.001) affected by growing-finishing diet and muscle type, with reduced cooking losses in the SW2 group for both muscles (Table 3). Seaweed addition significantly reduced shear force (*p <* 0.01), with SM having significantly (*p* < 0.001) lower shear force than LTL. No interaction effect between diet and muscle (*p* > 0.05) on quality properties of lamb meat was found.

The color measurements of meat exposed to air, chilled and stored for six days are shown in Table 4. Diet showed a significant effect on L* (lightness, *p* < 0.05), a* (redness, *p* < 0.001), and b* (yellowness, *p* < 0.05) values at Day 0. Additionally, diet showed a significant effect on a* (*p* < 0.001) and b* (*p* < 0.001) values at Day 3. Although SW1 showed higher a* values after three days of chilled storage compared to the other dietary treatments, this was not found on Day 6. On Day 6, diet affected only the b* value (*p* < 0.001). Metmyoglobin (MMb) content was significantly (*p* < 0.001) lower in SW1 meat then CON and SW2. Muscle type showed significant (*p* < 0.05) effects on L* and a* values during chilled storage. The interaction effect between diet and muscle on b* was significant both at Day 3 (*p* < 0.01) and 6 (*p* < 0.05, not shown).

Vacuum-packed LTL and SM+ADD from SW1 lambs showed significantly higher antioxidative activity (DPPH) on Day 0 (*p* < 0.01, not shown) and at the end of the chilled storage period (*p* < 0.01) relative to other dietary treatments (Figure 1A). At Day 0, the DPPH values were muscle dependent with SM+ADD having higher (*p* < 0.01) values than LTL, thus, significant (*p* < 0.05) interaction diet × muscle was found (not shown). No effects of muscle and interaction muscle × diet were observed (*p* > 0.05) after four weeks of storage. The thiobarbituric acid reactive substances (TBARS) content was significantly reduced (*p* < 0.001) in SM+ADD of lambs fed seaweed-supplemented diets relative to CON after four weeks storage (Figure 1B). In addition, an interaction between diet and muscle had a significant (*p* < 0.01) effect on TBARS content. CON meat had significantly (*p* < 0.001) higher warmed-over flavor (WOF) than the SW groups (Figure 1C). The WOF level was muscle dependent (*p* < 0.05), and the lowest WOF levels were measured in SM+ADD from the SW1 and SW2 group (*p* < 0.001). An interaction between diet and muscle had a significant (*p* < 0.001) effect on WOF.

### 3.3. Nutritive Profile of Meat: Fatty Acids and Minerals

The total SFA, MUFA, BCFA, and PUFA contents did not differ significantly (*p* > 0.1) among the diet groups. Thus, no significant difference among the diet groups was found for most fatty acids, as shown in Table 5. An effect of seaweed inclusion on the reduction of C18:1 n−5 content in SW lamb meat (*p* < 0.05) was found; i.e., a reduction relative to the CON meat. However, total PUFA and EPA (*p* < 0.05), LA (*p* < 0.01), DPA and DHA (*p* < 0.001) contents were significantly higher in the SM+ADD muscle than in the LTL muscle for all diets.

The seaweed-supplemented growing-finishing lamb diets had significant (*p* ≤ 0.001) effect on I and Se levels in lamb meat (Table 6). Iodine in plasma (Figure 2) increased correspondingly. The two seaweed-supplemented diets, SW1 and SW2 increased the level of I to 61.4 and 88.7 μg I/100 g of meat, respectively, relative to CON. The contents of Fe, Cu, and vitamin B12 in the meat were not affected (*p* > 0.05) by diet. Seaweed significantly (*p* < 0.001) increased As, while no differences (*p* > 0.1) in Cd contents were observed.

## 4. Discussion

The current feeding practice for Norwegian suckling lambs includes spring grazing on cultivated pasture and summer grazing on highland pasture. With poor weather conditions or lower slaughter weight (<42–45 kg live weight) lambs can be moved from the mountain pastures down to the farms and fed a diet consisting of concentrate and grass silage (3 to 6 weeks) until the slaughtering. Therefore, the fattening period presents an opportunity to increase carcass and meat quality traits and, thus, the nutritional value of lamb meat. The effect of seaweed inclusion rate in growing-finishing lamb diet on quality properties of meat has not been previously reported.

The iodine-rich seaweed supplemented at 2.5% and 5% dry matter (DM) considerably increased the iodine content in lamb diets (38 and 74 mg I/kg feed, respectively). However, supplementation of lamb diets with sundried *Saccharina latissima* during the last 35 days of the growing-finishing period, showed no effect on weight gain. Previously, Iannaccone et al. [18] reported upregulated cell growth pathways when sheep were supplemented with 10 mg I/kg feed indicating positive effect of iodine on animal growth. The relatively short-term feeding trial of the present study could not reveal positive effects of increased iodine intake on lamb carcass traits.

The key finding of the present study was the effect of the growing-finishing diet on meat characteristics, as no differences in the carcass traits in SW lamb relative to the control were found. An increase in pH with seaweed supplementation resulted in a positive effect on cooking loss and shear force. These results suggest that the antioxidants from sun-dried seaweed may have enhanced proteolytic activity in early postmortem muscle tissue. Similarly, Kannan et al. [19] reported that brown seaweed (Ascophyllum nodosum) extract-supplemented goat diets can enhance the antioxidative status of an animal. Shear force values for LTL muscle from lambs fed seaweed-supplemented diets and all SM muscles were below the threshold level (33.34 N/cm^2^) for tender lamb [20].

The color of lamb meat is an important trait for consumer acceptance; a* = 14.8 was defined as an acceptable threshold for meat redness [21]. In the present study, lamb meat from all dietary groups on Day 3 and Day 6 had slightly lower a* values then the recommended threshold. In addition, a consumer’s perception of meat redness is difficult to predict using objective measurements due to variability in consumer data [22]. Furthermore, SW1 had an overall positive effect on the color stability of both examined muscles due to the increased a* (redness) on Day 0 and 3, thus, reduced b* (yellowness) values and a lower percentage of metmyoglobin (MMb) as measured after six days of chilled storage, relative to CON and SW2. Additionally, the reduction of MMb content is considered to improve color stability. Higher oxidative activity (DPPH) of SW1 lamb meat after six days of chilled storage delayed formation of MMb. Reduced MMb content was also reported in goat meat, when an extract from brown seaweed (Ascophyllum nodosum) was added to a goat diet [23]. In contrast, the present study showed increased formation of MMb in SW2 meat during storage, when seaweed was supplemented at 5% DM.

The antioxidant capacity of a muscle is influenced by antioxidants in seaweed, but also by the content of available minerals (Fe, Se, Zn, Cu, Mn) in the muscle [24]. Brown seaweed has shown its ability to alter antioxidant activity in ruminants [25]. However, the present study indicates that *S. latissima* at 5% DM induced an imbalance between antioxidants and oxidants in lamb muscles, with a shift towards oxidative processes. However, the lipid oxidation (TBARS) levels in lamb meat after four weeks of chilled storage were still below the lipid oxidation threshold for off-flavor (<0.5 mg MDA/kg of meat) [26]. Additionally, the results for warmed-over flavor (WOF) in SW1 and SW2 SM+ADD were below, while all LTL and CON SM+ADD were above, the off-flavor threshold. The observed differences in lipid oxidation level may also be related to variations in muscle-type metabolism.

The seaweed-supplemented growing-finishing lamb diet had a minor effect on fatty acid composition of lamb meat. This result might be due to a relatively low lipid content in *S. latissima* (0.7% of DM); thus, the seaweed contributed with only 0.7 and 1.2% of the lipids in the SW1 and SW2 diets, respectively. Although fatty acid composition of lamb meat is associated with both dietary fats and hydrogenation of ingested fats by rumen bacteria, supplementation of seaweed for 35 days has not induced changes.

As previously reported, the level of I in the SM+ADD muscle of lamb fed with the seaweed-supplemented diet (5% DM) increased more than 20 times [7]. The present study has shown that the LTL muscle has an even higher iodine (I) bioaccumulation potential: the content of I in LTL was 26 times higher in lambs fed SW1 and 38 times higher in lambs fed SW2 compared with the CON diet. Surprisingly, selenium (Se) content in SW meat increased, although, the level of Se in growing-finishing lamb diet had slightly decreased with the substitution of barely with seaweed (Table 1). As previously reported by Gómez-Jacinto et al. [27], in in vitro digestion Se from algae has higher bioaccessibility. Specifically, they found enhanced bioavailability of Se in mice tissues when fed Se-enriched algae diets [27]. This aligns with the results in a present study where the Se-specie in *S. latissima* could possibly have higher bioavailability than in barely. Higher Se content in SW1 compared to the SW2 diet showed no effect on Se levels in LTL muscle. In general, stabilization of Se retention in lamb muscle has co-occurred with seaweed supplementation.

The two seaweed-supplemented diets, SW1 and SW2, increased the level of I to 61.4 and 88.7 μg I/100 g of meat, respectively, relative to CON (2.34 μg I/100 g of meat). Considering an Adequate Intake (AI) of 150 μg I/day for adults [28], CON meat can provide only 1.56% of AI. Meat from SW1 and SW2 lambs can provide on average 40.9% and 59.1% of the AI for I, respectively. The Se content in CON LTL was on average 13.86 μg Se/100 g, while the average Se content in SW1 and SW2 was 15.64 μg Se/100 g meat. In addition, EFSA set an adequate intake (AI) level of 70 μg Se/day for adults [28]. On average, 100 g of meat from CON- and SW-fed lambs provided 19.8% and 22.34% of the AI, respectively. Furthermore, foodstuffs containing >15% of the Recommended Daily Allowance (RDA) can obtain nutrition labeling as a “source of nutrient” [29]. Considering that the RDA for Se and I are 55 μg/day and 150 μg/day, respectively, SW lamb meat with the lowest Se and I content (26% and 28% RDA, respectively) would still meet the requirement. Meat containing more than 22.5 μg I/100 g of meat (15% of RDA) may use the health claim label “contributes to the normal production of thyroid hormones, thyroid function, normal energy yielding metabolism, and maintenance of normal skin” [30].

On the other hand, supplementation of seaweed also increased the arsenic (As) content, while having no effect on cadmium (Cd) content in SW meat. Although, with higher SW inclusion levels, the Cd content has increased in growing-finishing diets that had no effect on Cd content in lamb meat. The Cd content found in SW lamb meat was far below the maximum allowed level of 5 µg Cd/100 g of meat [31]. Based on the obtained results for the total As in lamb meat, the inorganic As potentially presents 0.015 and 0.03 µg/100 g meat in SW1 and SW2, respectively. Considering the EC acceptance level for inorganic As in rice and rice-based products with 10–30 µg/100 g [32], an estimated inorganic As content was far below that threshold level. Nevertheless, the increase in total As content in lamb meat after 35 days of feeding a seaweed-supplemented diet suggested a need for optimizing seaweed inclusion levels, the duration of the finishing feeding or seaweed processing to reduce the As level. In addition, more research is needed to understand the relationship between phenotype and biological pathways involved in muscle microelements deposition.

## 5. Conclusions

Supplementing growing-finishing lamb with *Saccharina latissima* had overall positive effects on the cooking loss and shear force, color stability and oxidation level during chilled storage. In addition, seaweed supplementation can enrich lamb meat with iodine and help stabilization of the selenium content. However, some challenges should be overcome, such as a high iodine level in animal feed and an increased arsenic content in lamb meat. Therefore, it is necessary to optimize sun-dried seaweed inclusion levels and the length of finishing feeding. Further studies should evaluate administration of other types of seaweeds on lamb meat quality and mineral content.

## Figures and Tables

**Figure 1 foods-12-02131-f001:**
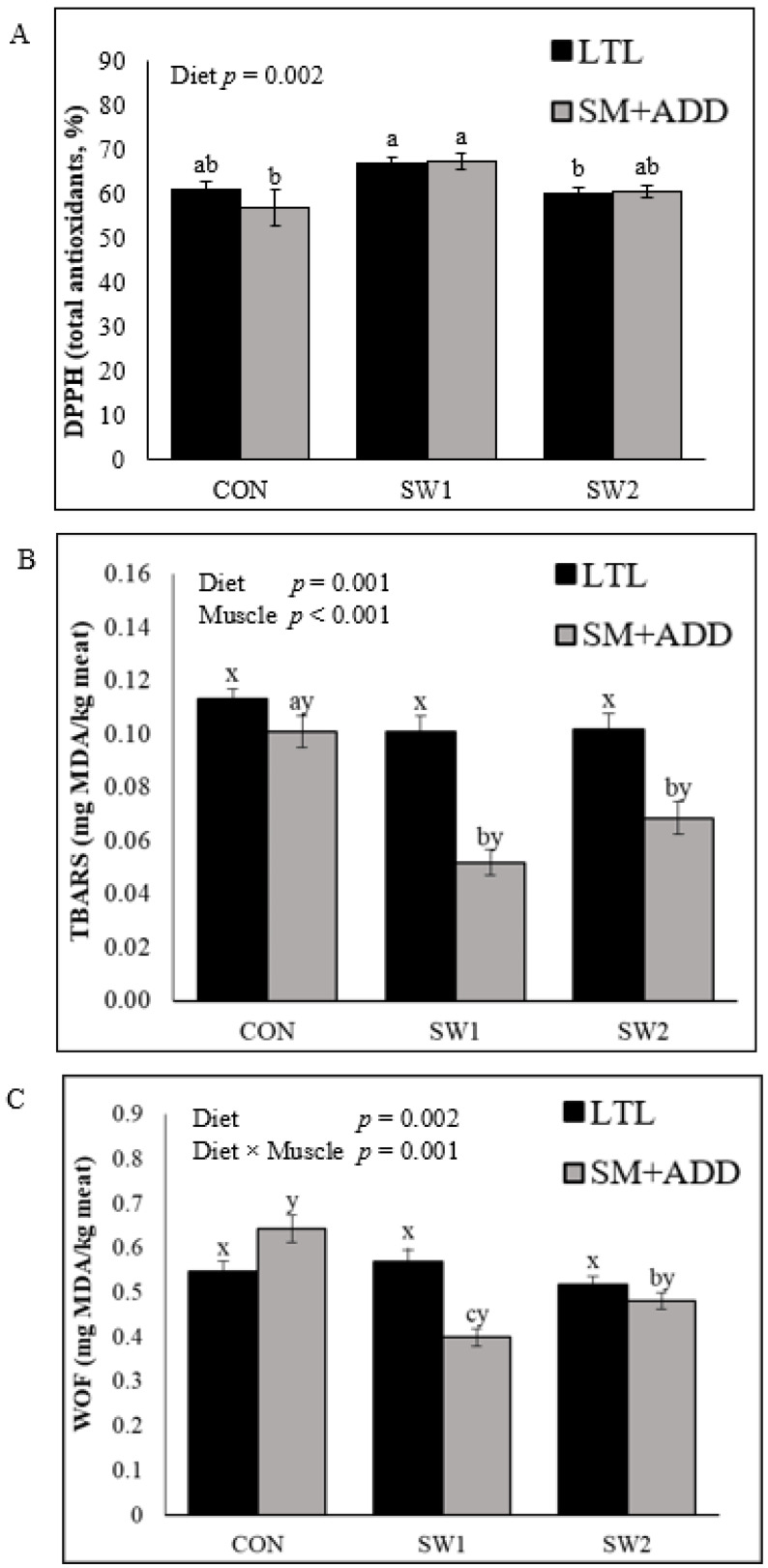
Effect of growing-finishing diet on longissimus thoracis et lumborum (LTL) and semimembranosus with adductor (SM+ADD) vacuum packed (**A**) total antioxidant capacity (DPPH) and (**B**) oxidative stability (TBARS) after four weeks of chilled storage, and (**C**) warm-over flavor (WOF). CON = control diet; SW1 and SW2 = control + seaweed inclusion level of 2.5% and 5% DM, respectively. Error bars represent the standard error of means (*n* = 8). ^a–c^ Means with different subscripts within muscle and diet groups were significantly different (*p* < 0.05). ^x,y^ Means with different subscripts within a muscle were significantly different (*p* < 0.05).

**Figure 2 foods-12-02131-f002:**
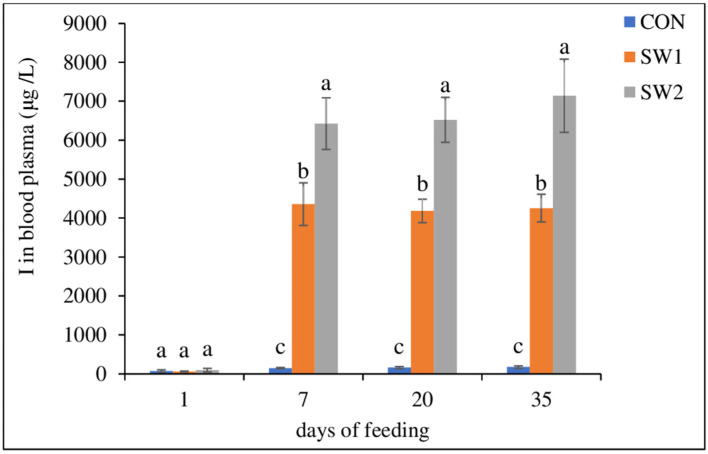
The I content in the blood plasma of lamb (*n* = 24) affected by dietary treatments during 35 days of the growing-finishing period. CON = control diet; SW1 and SW2 = control + seaweed inclusion level of 2.5% and 5% DM, respectively. Error bars represent standard error of means (*n* = 8). ^a–c^ Means with different subscripts within the same day of feeding are significantly different (*p* < 0.05).

**Table 1 foods-12-02131-t001:** Ingredient (g/kg) and chemical composition of growing-finishing lamb feed ^1^.

Ingredients	CON ^a^	SW1 ^2^	SW2 ^b^
Early cut grass/clover silage	822.0	822.0	822.0
Wilted seaweed ^3^	-	33.9	67.9
Compound feed (DRØV lam) ^4^	102.2	102.2	102.2
Rolled barley	21.6	10.8	-
VitaMineral^®^ Normal Sau ^5^	4.8	4.8	4.8
GrassAAT Korn ^6^	3.1	3.1	3.1
Added free water	46.3	23.2	-
Analyzed content, g/kg DM			
Dry matter, g/kg fresh feed	364.1	361.3	358.4
Crude protein	190.3	189	186.4
Neutral detergent fiber, NDF	453.5	451.0	433.4
Acid detergent fiber, ADF	282	281.4	279.1
Ash	91.9	99.5	105.6
Starch	73.4	68.8	61.3
Gross energy (MJ/kg DM)	19.2	19.0	18.9
Analyzed content of minerals, mg/kg DM feed			
Fe	0.48	0.53	0.51
I	5.40	105.50	204.60
Se	0.40	0.37	0.35
Co	6.77	6.70	6.73
As	0.14	1.83	3.66
Cd	0.03	0.06	0.09
Fatty acids, g/kg DM feed			
C16:0	5.45	4.55	5.37
C18:0	0.46	0.41	0.44
C18:1 n−9	3.29	3.01	3.87
C18:2 n−6	6.46	5.61	6.96
C18:3 n−3	9.98	7.89	9.44

^a,b^ Ingredients, analyzed content, and minerals of CON and SW2 diets were given by Grabež et al. [7]. ^1^ CON = control diet; SW1 and SW2 = control + seaweed inclusion level of 2.5% and 5% DM, respectively; ^2^ Calculated ingredient composition of the other two diets, while chemical composition is based on analysis of the diet itself; ^3^ Wilted seaweed with DM content of 283 g/kg, and 118, 373, 234, 409 g/kg DM of CP, aNDFom, aADFom, ash in respective order and gross energy value of 10.7 MJ/kg DM; ^4^ Commercial compound feed produced and supplied by Norgesfôr AS (Mysen, Norway), with CP, aNDFom and crude fat contents of 153, 235, and 43 g/kg, respectively; ^5^ Vitamin and mineral supplement for sheep (Vilomix Norway AS; Hønefoss, Norway) containing vitamins (A 100 IE/g; D3 100 IE/g; and E 2000 mg/kg), macro-minerals (g/kg of Ca 140; P 70; Mg 60; Na 90 and S 10), and micro-minerals (mg/kg of Mn 3000; Zn 5000; Co 30; I 100 and Se 25); ^6^ GrassAAT Korn is a preservative and feed stabilizer.

**Table 2 foods-12-02131-t002:** Effect of growing-finishing diet on weight gain (ADWG) and carcass traits.

Item	CON ^1^	SW1	SW2	SEM ^2^	*p*-Value
ADWG ^3^ (g/day)	319.0	323.6	311.4	8.49	0.844
Dressing percentage	41.8	40.7	41.3	0.35	0.483
CCW ^4^ (kg)	20.30	19.85	19.91	0.28	0.782
EU conformation ^5^	9.25 (R+)	8.88 (R)	9.75 (R+)	0.15	0.089
EU fatness ^6^	7.75 (3−)	7.50 (3−)	7.75 (3−)	0.22	0.870

^1^ CON = control diet; SW1 and SW2 = control + seaweed inclusion level of 2.5% and 5% DM, respectively; ^2^ SEM = pooled standard error of mean; ^3^ ADWG = average daily weight gain; ^4^ CCW = cold carcass weight; ^5^ Scale 1–15 points: 1 = P−; 2 = P (poor); 3 = P+; 4=O−; 5=O (normal); 6 = O+; 7 = R−; 8 = R (good); 9 = R+; 10 = U−; 11=U (very good); 12 = U+; 13 = E−; 14 = E (excellent); 15 = E+; ^6^ Scale 1–15 points: 1 = 1−; 2=1 (very scarce); 3 = 1+; 4 = 2−; 5 = 2 (scarce); 6 = 2+; 7 = 3−; 8 = 3 (medium); 9 = 3+; 10 = 4−; 11 = 4 (important); 12 = 4+; 13 = 5−; 14 = 5 (excellent); 15 = 5+.

**Table 3 foods-12-02131-t003:** Effect of growing-finishing diet on quality properties of lamb meat.

Items	Diet ^1^			SEM ^2^	Muscle ^3^		SEM	*p*-Value	
	CON	SW1	SW2		LTL	SM		Diet	Muscle
pH	5.54 ^b^	5.63 ^a^	5.47 ^c^	0.01	5.55	5.54	0.01	<0.001	0.668
Cooking losses (%)	22.94 ^a^	23.37 ^a^	19.44 ^b^	0.25	22.59 ^a^	20.99 ^b^	0.36	<0.001	0.001
SF (N/cm^2^) ^4^	33.12 ^a^	28.30 ^ab^	26.97 ^b^	1.03	32.29 ^a^	26.34 ^b^	0.99	0.006	<0.001

^1^ CON = control diet; SW1 and SW2 = control + seaweed inclusion level of 2.5% and 5% DM, respectively; ^2^ SEM = pooled standard error of mean; ^3^ LTL = longissimus thoracis et lumborum, SM = semimembranosus; ^4^ SF = shear force; ^a–c^ Means with different subscripts are significantly different (*p* < 0.05).

**Table 4 foods-12-02131-t004:** Effect of growing-finishing diet on meat surface color during chilled-storage under aerobic conditions.

Item ^1^	Day	LTL ^2^			SM			SEM ^4^	*p*-Value		
		CON ^3^	SW1	SW2	CON	SW1	SW2		Diet	Muscle	Diet × Muscle
L*	0	41.59 ^a^	40.47 ^b^	40.83 ^a^	40.86 ^a^	39.98 ^b^	41.03 ^a^	0.33	0.002	0.152	0.249
	3	42.19 ^A^	42.50 ^A^	42.65 ^A^	41.46 ^B^	40.85 ^B^	41.77 ^B^	0.27	0.071	<0.001	0.127
	6	43.20 ^A^	43.13 ^A^	42.98 ^A^	41.98 ^B^	41.25 ^B^	42.04 ^B^	0.34	0.179	<0.001	0.506
a*	0	8.13 ^bB^	8.99 ^aB^	8.11 ^bB^	8.77 ^bA^	9.41 ^aA^	9.00 ^bA^	0.17	<0.001	<0.001	0.279
	3	12.38 ^b^	13.00 ^a^	12.04 ^b^	12.19 ^b^	13.42 ^a^	12.48 ^b^	0.26	<0.001	0.226	0.293
	6	15.51 ^B^	12.79 ^B^	12.53 ^B^	13.05 ^A^	12.89 ^A^	12.98 ^A^	0.24	0.917	0.034	0.526
b*	0	9.15 ^ab^	9.30 ^a^	8.68 ^b^	9.14 ^ab^	9.22 ^a^	9.01 ^b^	0.17	0.013	0.506	0.331
	3	12.89 ^b^	13.35 ^a^	12.79 ^a^	12.45 ^b^	13.50 ^a^	13.47 ^a^	0.19	<0.001	0.347	0.005
	6	13.57 ^a^	13.05 ^b^	13.48 ^a^	13.86 ^a^	12.55 ^b^	13.64 ^a^	0.19	<0.001	0.904	0.042
MMb (%)		26.54 ^ab^	22.38 ^b^	30.96 ^a^	27.69 ^ab^	23.23 ^b^	29.88 ^a^	1.18	<0.001	0.855	0.840

^1^ L* = lightness, a* = redness, b* = yellowness, MMb = percentage of metmyoglobin; ^2^ LTL = longissimus thoracis et lumborum, SM = semimembranosus; ^3^ CON = control diet; SW1 and SW2 = control + seaweed inclusion level of 2.5% and 5% DM, respectively; ^4^ SEM = pooled standard error of mean; ^a,b^ Means with different subscripts indicate significant difference (*p* < 0.05) between diets; ^A,B^ Means with different subscripts indicate significant difference (*p* < 0.05) between muscle type.

**Table 5 foods-12-02131-t005:** Effect of growing-finishing diet on the fatty acids in lamb meat (mg/100 g meat).

Items	Diet ^1^			Muscle ^2^		SEM ^3^	*p*-Value	
	CON	SW1	SW2	LTL	SM+ADD		Diet	Muscle
C10:0	2.64	2.42	2.66	2.69	2.45	0.30	0.936	0.690
C12:0	3.78	4.75	4.52	3.84	4.86	0.55	0.756	0.360
C14:0	52.0	59.4	59.12	56.57	57.11	6.51	0.873	0.967
C15:0	7.92	8.85	8.60	7.79	9.13	0.94	0.916	0.480
C16:0	460.2	487.9	492.8	493.5	467.2	39.50	0.936	0.741
C17:0	20.82	19.62	19.83	19.63	20.55	1.90	0.963	0.809
C18:0	341.4	349.3	338.0	353.3	332.6	28.24	0.986	0.716
C20:0	1.64	1.66	2.11	1.96	1.64	0.24	0.671	0.511
C23:0	2.07	1.91	2.24	1.80 ^b^	2.35 ^a^	0.10	0.401	0.009
iso–C15:0	2.75	2.78	2.77	2.79	2.74	0.36	0.990	0.934
anteiso–C15:0	4.47	2.27	5.10	4.56	5.33	0.42	0.720	0.364
iso–C16:0	3.16	3.40	3.21	3.18	3.33	0.40	0.960	0.843
iso–C17:0	9.82	10.33	10.16	9.45	10.76	0.74	0.960	0.379
iso–C18:0	2.47	2.35	2.21	2.44	2.25	0.28	0.930	0.732
C14:1 n−5	1.50	1.76	1.86	1.55	1.87	0.23	0.817	0.494
C16:1 n−9	6.05	6.87	6.41	5.98	6.90	0.65	0.877	0.482
C16:1 n−7	218.58	215.5	219.24	220.66	214.9	5.33	0.955	0.592
C17:1 n−7	8.08	8.36	7.85	7.70	8.50	0.33	0.820	0.235
C18:1 n−9t	1.80	1.88	2.0	1.82	1.97	0.17	0.890	0.651
C18:1 n−8t	6.44	6.05	6.57	6.33	6.38	0.44	0.883	0.954
C18:1 n−7t	32.02	32.89	31.47	31.55	32.71	3.69	0.988	0.876
C18:1 n−9	790	752	738.7	742.4	778.0	65.09	0.947	0.786
C18:1 n−7	20.89	20.60	21.49	19.72	22.27	1.50	0.970	0.400
C18:1 n−5	1.89 ^a^	1.15 ^b^	1.43 ^ab^	1.59	1.39	0.11	0.037	0.389
C18:2 n−6 t9,12	5.15	5.16	5.07	4.78	5.47	0.49	0.997	0.486
C18:2 n−6 (LA)	78.95	82.45	84.88	73.86 ^b^	90.33 ^a^	2.80	0.687	0.005
C18:3 n−6	0.68	0.67	1.35	1.04	0.75	0.24	0.413	0.540
C18:3 n−3 (ALA)	29.98	29.36	31.46	27.81	32.73	1.45	0.834	0.098
C18:2 c9,t11 (CLA)	14.18	17.41	15.98	14.09	17.63	1.77	0.758	0.324
C20:2 n−6	0.42	0.41	0.70	0.57	0.45	0.10	0.408	0.565
C20:3 n−6	3.52	3.57	3.46	3.21 ^b^	3.82 ^a^	0.10	0.923	0.005
C20:4 n−6	28.77	32.58	29.59	26.74 ^b^	33.89 ^a^	0.82	0.151	<0.001
C20:5 n−3 (EPA)	21.24	20.90	22.16	19.88 ^b^	22.99 ^a^	0.58	0.658	0.010
C22:4 n−6	1.24	1.43	1.69	1.44	1.47	0.13	0.379	0.897
C22:5 n−3 (DPA)	21.11	22.64	22.45	20.00 ^b^	24.14 ^a^	0.56	0.440	<0.001
C22:6 n−3 (DHA)	7.35	6.87	6.86	5.92 ^b^	8.13 ^a^	0.28	0.719	<0.001
∑SFA	893	936	929.9	941.0	898.0	77.49	0.970	0.782
∑BCFA	22.66	24.12	23.44	22.42	24.40	2.13	0.961	0.643
∑MUFA	1087	1047	1037	1039.3	1075	73.40	0.958	0.810
∑PUFA	212.6	223.5	225.7	199.3 ^b^	241.8 ^a^	8.06	0.779	0.012
∑PUFA n−6	113.58	121.11	121.68	106.86 ^b^	130.72 ^a^	3.71	0.614	0.003
∑PUFA n−3	79.68	79.78	82.92	73.60 ^b^	87.99 ^a^	2.62	0.848	0.009
∑n−6/∑n−3	1.43	1.51	1.47	1.45	1.49	0.02	0.154	0.232
∑PUFA/∑SFA	0.29	0.29	0.27	0.24 ^b^	0.33 ^a^	0.01	0.740	0.002

^1^ CON = control diet; SW1 and SW2 = control + seaweed inclusion level of 2.5% and 5% DM, respectively; ^2^ LTL = longissimus thoracis et lumborum, SM+ADD = semimembranosus + adductor; ^3^ SEM = pooled standard error of the mean; ^a,b^ Means with different subscripts are significantly different (*p* < 0.05). ∑SFA = sum of saturated fatty acids, ∑BCFA = sum of branch chained fatty acids, ∑MUFA = sum of monounsaturated fatty acids, PUFA = sum of polyunsaturated fatty acids, LA = linoleic acid, ALA = α-linolenic acid, CLA = conjugated linoleic acid, EPA = eicosapentanoic acid, DPA = docosapentanoic acid, DHA = docosahexanoic acid, ∑n−3 = sum of omega-3 fatty acids, ∑n−6 = sum of omega-6 fatty acids, ∑n−6/∑n−3 = ratio of sum omega-6 o sum omega-3 fatty acid; ∑PUFA/∑SFA = ratio of sum PUFA and sum SFA.

**Table 6 foods-12-02131-t006:** The micronutrient and contaminant content (μg/100 g of meat) in lamb meat.

Item ^1^	CON ^2^	SW1	SW2	SEM ^3^	*p*-Value
Fe (mg/100 g)	2.61	2.66	2.75	0.13	0.430
	(2.4–2.9) ^4^	(2.3–3.1)	(2.4–3.1)		
Se	13.86 ^b^	15.64 ^a^	15.64 ^a^	0.51	0.001
	(12.0–15.4)	(14.3–17.1)	(14.9–16.0)		
Cu	163.6	162.1	165.7	0.01	0.856
	(142.9–188.6)	(148.6–171.4)	(148.6–182.9)		
I	2.34 ^c^	61.4 ^b^	88.7 ^a^	5.02	<0.001
	(2.1–3.1)	(41.7–74.4)	(71.7–103.0)		
As	0.23 ^c^	1.54 ^b^	3.09 ^a^	0.20	<0.001
	(0.1–1.1)	(1.1–2.0)	(2.6–3.7)		
Cd	0.02	0.03	0.02	0.01	0.244
	(0.01–0.04)	(0.02–0.05)	(0.015–0.02)		
Vitamin B_12_	0.90	0.85	0.92	0.09	0.668
	(0.8–1.7)	(0.7–1.0)	(0.6–1.2)		

^1^ Content measured in longissimus thoracis et lumborum muscle; ^2^ CON = control diet; SW1 and SW2 = control + seaweed inclusion level of 2.5% and 5% DM, respectively; ^3^ SEM = pooled standard error of the mean; ^4^ min–max level; ^a–c^ Means with different subscripts are significantly different (*p* < 0.05).

## Data Availability

The data presented in this study are available on request from the corresponding author.

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
