# Peer review of "Sugar Kelp (Saccharina latissima) Seaweed Added to a Growing-Finishing Lamb Diet Has a Positive Effect on Quality Traits and on Mineral Content of Meat"

_foods, 2023, doi:10.3390/foods12112131_

Round 1
Reviewer 1 Report
The aim of the study was to to evaluate the effect of in creasing levels of the brown seaweed Saccharina latissima in growing-finishing lamb diets on quality traits of fresh meat. The experiment has been well planed. The Materials and Methods section is very precise. The results have been clear presented and well disscused. The conclusions from the research were correctly formulated. In my opinion, the manuscript should consider minor revisions:
Line 99: Please insert „Dry matter” before „(DM)”. Please explain the abbreviations used the first time throughout the manuscript.
Line 117: Please insert „value” after „(pH)”.
Line 128 and Lines 137-147: Please note that shear force is not an organoleptic characteristic of meat. This measurement cannot be treated as meat tenderness.
Lines 142-142: Meat samples (1 × 1 × 4 cm) were cut along the fiber direction from slice thickness 2 cm. Please explain how this was possible. How many samples were cut from a slice of meat?
Table 5: „Fe (mg/100g)”. Please insert a space between a value and a unit.
Author Response
Dear Editor,
We hereby submit the revised manuscript (Manuscript ID: foods-2407622) entitled “Sugar kelp (Saccharina latissima) seaweed added to a growing-finishing lamb diet has a positive effect on quality traits and on mineral content of meat”. The manuscript has been carefully revised taking into consideration Editors’, Reviewer #1, #2 and #3 suggestions and comments. We thank all for very useful comments.
Thank you very much for considering our manuscript. We are looking forward to your response.
With respect,
Vladana Grabež & coauthors
Editor
- Please revise your manuscript according to the referees’ comments and upload the revised file within 5 days.
- Please use the version of your manuscript found at the above link for your revisions.
- Please check that all references are relevant to the contents of the manuscript.
- Any revisions made to the manuscript should be marked up using the “Track Changes” function if you are using MS Word/LaTeX, such that changes can be easily viewed by the editors and reviewers.
- Please provide a short cover letter detailing your changes for the editors’ and referees’ approval.
If one of the referees has suggested that your manuscript should undergo extensive English revisions, please address this issue during revision. We propose that you use one of the editing services listed at https://www.mdpi.com/authors/english or have your manuscript checked by a colleague fluent in English writing.
Detailed response to the Editor:
The revision of Manuscript was done according to Editor’s instructions and all comments provided by Reviewers were taken into consideration and detailed improvements were done.
Reviewer #1:
The aim of the study was to to evaluate the effect of increasing levels of the brown seaweed Saccharina latissima in growing-finishing lamb diets on quality traits of fresh meat. The experiment has been well planed. The Materials and Methods section is very precise. The results have been clear presented and well disscused. The conclusions from the research were correctly formulated.
Response:
The authors thank to Reviewer #1 for a nice comments.
Reviewer #1:
In my opinion, the manuscript should consider minor revisions:
Line 99: Please insert „Dry matter” before „(DM)”. Please explain the abbreviations used the first time throughout the manuscript.
Response:
In agreement with Reviewer #1 comment authors made correction in Line 85-86.
Reviewer #1:
Line 117: Please insert „value” after „(pH)”.
Response:
In agreement with Reviewer #1 comment authors made correction in Line 156.
Reviewer #1:
Line 128 and Lines 137-147: Please note that shear force is not an organoleptic characteristic of meat. This measurement cannot be treated as meat tenderness.
Response:
In agreement with Reviewer #1 comment authors made correction in Line 167 and Line 177-193.
Reviewer #1:
Lines 142-142: Meat samples (1 × 1 × 4 cm) were cut along the fiber direction from slice thickness 2 cm. Please explain how this was possible. How many samples were cut from a slice of meat?
Response:
Authors have revised sub-section 2.4 and in Line 188- 193 is written: Warner–Bratzler shear force (SF) analyses were performed using a Texture Analyzer (Model: TA-HDi, Producer: Stable Micro Systems, Godalming, UK) equipped with a shear cell HDP/BSK and fitted with a 25 kg load cell and shear measurements done perpendicular to the muscle fiber direction. The samples were tempered to room temperature and 6-10 cylindrical samples (1 × 1 × 4 cm ) were cut along the fiber direction, avoiding fat and connective tissue.
Reviewer #1:
Table 5: „Fe (mg/100g)”. Please insert a space between a value and a unit.
Response:
In agreement with Reviewer #1 comment authors made correction in Table 5: Fe (mg/100 g).

Reviewer 2 Report
The authors evaluated the effects of different levels of brown seaweed diets on lamb meat quality, including the concentrations of desirable and toxic elements. The manuscript is interesting to the scientific community because currently there is a link between nutrition and health and the use of different natural substances. The authors placed their study in a broad context in Introduction section, however, they did not highlight the purpose of the study in Introduction and Abstract.
Other comments:
Abbreviations should be defined the first time they appear in each of the sections: the abstract; the main text; the first figure or table. However, DPPH (line 132), TBARS, WOF, FAME (line134) were not defined. Please, check throughout the manuscript.
Lines 148-149 How was the cooking loss calculated? How was juice detected after meat cooking? Or it was the ratio between the sample weight before and after cooking?
Why is the p-value denoted as p<0.01 in one case (line 308) and p=0.004 in the other case (line 312)? I would suggest to unify everywhere by using either p<0.05; p<0.01 and p<0.001 or fixed exact p-values (p=0.0xx).
Tables and Figures should be inserted into the main text close to their first citation and must be numbered following their number of appearance (Table 1, Table 2, Table 3). In the manuscript this provision is not followed. It should be revised. I would like to suggest to move Table 1 presenting feed ingredients and chemical composition to Materials and Methods section.
Author Response
Dear Editor,
We hereby submit the revised manuscript (Manuscript ID: foods-2407622) entitled “Sugar kelp (Saccharina latissima) seaweed added to a growing-finishing lamb diet has a positive effect on quality traits and on mineral content of meat”. The manuscript has been carefully revised taking into consideration Editors’, Reviewer #1, #2 and #3 suggestions and comments. We thank all for very useful comments.
Thank you very much for considering our manuscript. We are looking forward to your response.
With respect,
Vladana Grabež & coauthors
Editor
- Please revise your manuscript according to the referees’ comments and upload the revised file within 5 days.
- Please use the version of your manuscript found at the above link for your revisions.
- Please check that all references are relevant to the contents of the manuscript.
- Any revisions made to the manuscript should be marked up using the “Track Changes” function if you are using MS Word/LaTeX, such that changes can be easily viewed by the editors and reviewers.
- Please provide a short cover letter detailing your changes for the editors’ and referees’ approval.
If one of the referees has suggested that your manuscript should undergo extensive English revisions, please address this issue during revision. We propose that you use one of the editing services listed at https://www.mdpi.com/authors/english or have your manuscript checked by a colleague fluent in English writing.
Detailed response to the Editor:
The revision of Manuscript was done according to Editor’s instructions and all comments provided by Reviewers were taken into consideration and detailed improvements were done.
Reviewer #2:
The authors evaluated the effects of different levels of brown seaweed diets on lamb meat quality, including the concentrations of desirable and toxic elements. The manuscript is interesting to the scientific community because currently there is a link between nutrition and health and the use of different natural substances. The authors placed their study in a broad context in Introduction section, however, they did not highlight the purpose of the study in Introduction and Abstract.
Response:
Line 18-20: The objective of the present study was to investigate the use of Saccharina latissima in a lamb diet to improve the eating quality and nutritional value of meat.
Line 73-75: Therefore, the present study evaluated a feeding strategy, using increasing levels of the brown seaweed Saccharina latissima in growing-finishing lamb diets, in order to enhance quality traits of fresh meat
Reviewer #2:
Other comments:
Abbreviations should be defined the first time they appear in each of the sections: the abstract; the main text; the first figure or table. However, DPPH (line 132), TBARS, WOF, FAME (line134) were not defined. Please, check throughout the manuscript.
Response:
In agreement with Reviewer #2 comment authors made corrections through the Manuscript.
Reviewer #2:
Lines 148-149 How was the cooking loss calculated? How was juice detected after meat cooking? Or it was the ratio between the sample weight before and after cooking?
Response:
In agreement with Reviewer #2 comment authors made revision of Sub-section “2.4. Cooking loss, shear force and color” in Line 179-193:
Warner–Bratzler shear force (SF) analyses were performed on LTL and SM samples on Day 7 (9 days post-mortem). Vacuum-packed meat (slice thickness 2 cm) LTL and SM samples (slice thickness cca 2 cm) on Day 7 (9 days post-mortem) was were placed in a preheated water bath at 80℃ and cooked until reaching an internal temperature of 72℃. After cooking, the samples were cooled in ice-cold water to 20℃. The next day, cooked meat was removed from the plastic bag, dried with a paper towel and weight (W1). Liquid loss of cooked meat was weight (W2). meat samples (1 × 1 × 4 cm) were cut along the fiber direction, avoiding fat and connective tissue. Shear force analyses were performed using a Texture Analyzer (Model: TA-HDi, Producer: Stable Micro Systems, Godalming, UK) equipped with a shear cell HDP/BSK and fitted with a 25 kg load cell, taking 10 shear measurements perpendicular to the muscle fiber direction.
Cooking loss was calculated with the following calculation:as the ratio between the weight of juice after cooking and the total weight of the cooked sample.
Cooking loss (%)= [W2⁄((W1+W2))]×100
Warner–Bratzler shear force (SF) analyses were performed using a Texture Analyzer (Model: TA-HDi, Producer: Stable Micro Systems, Godalming, UK) equipped with a shear cell HDP/BSK and fitted with a 25 kg load cell. Shear measurements were done perpendicular to the muscle fiber direction. onThe samples were equilibrated to room temperature and 6-10 LTL and SM samples on Day 7 (9 days post-mortem).cylindrical samples (1 × 1 × 4 cm ) were cut along the fiber direction, avoiding fat and connective tissue.
Reviewer #2:
Why is the p-value denoted as p<0.01 in one case (line 308) and p=0.004 in the other case (line 312)? I would suggest to unify everywhere by using either p<0.05; p<0.01 and p<0.001 or fixed exact p-values (p=0.0xx).
Response:
In agreement with Reviewer #2 comment authors made corrections through the Result section.
Reviewer #2:
Tables and Figures should be inserted into the main text close to their first citation and must be numbered following their number of appearance (Table 1, Table 2, Table 3). In the manuscript this provision is not followed. It should be revised. I would like to suggest to move Table 1 presenting feed ingredients and chemical composition to Materials and Methods section.
Response:
In agreement with Reviewer #2 comment authors moved Tables and Figures close to their first citation.

Reviewer 3 Report
In general, all parts of the article are written in understandable language and the transitions between sections are well organized. I would like to congratulate the authors for the good quality of the article, the literature reported used to write the paper, and for the clear and appropriate structure. The manuscript is well written, presented and discussed, and understandable to a specialist readership. The organization and the structure of the article are satisfactory and in agreement with the journal instructions for authors. The subject is adequate with the overall journal scope.
Here are some suggestions I made to improve the article;
1. In order to further improve the quality of the paper, an overall check of the English language is recommended.
2. It is understood from the results that diet x muscle interactions are investigated, but analysis results are not given in the tables. It would be better if the P values of the interactions were added to the tables.
3. Abbreviations given in the tables should be explained under the table, for example SF in Table 3.
4. Why are the TBARS, DPPH, L, a, b and MMB values not given in the table? It should be better added to Table 3.
5. Figure 2, graph c has written TBARS instead of WOF, check and correct it.
6. Only the important interactions should be shown in the figures, there is no need to give the unimportant ones. Just giving the values in the table is enough (eg DPPH).
7. In the discussion, chilled storage was mentioned for color and TBARS values, but we cannot see the data related to this storage in Tables or figures. It would be nice if these results were added.
8. The discussion is insufficient, you can improve this section with previous studies in this area.
9. I could not see the animal experiments ethics committee approval number in the article. It must be added.
In order to further improve the quality of the paper, an overall check of the English language is recommended.
Author Response
Dear Editor,
We hereby submit the revised manuscript (Manuscript ID: foods-2407622) entitled “Sugar kelp (Saccharina latissima) seaweed added to a growing-finishing lamb diet has a positive effect on quality traits and on mineral content of meat”. The manuscript has been carefully revised taking into consideration Editors’, Reviewer #1, #2 and #3 suggestions and comments. We thank all for very useful comments.
Thank you very much for considering our manuscript. We are looking forward to your response.
With respect,
Vladana Grabež & coauthors
Editor
- Please revise your manuscript according to the referees’ comments and upload the revised file within 5 days.
- Please use the version of your manuscript found at the above link for your revisions.
- Please check that all references are relevant to the contents of the manuscript.
- Any revisions made to the manuscript should be marked up using the “Track Changes” function if you are using MS Word/LaTeX, such that changes can be easily viewed by the editors and reviewers.
- Please provide a short cover letter detailing your changes for the editors’ and referees’ approval.
If one of the referees has suggested that your manuscript should undergo extensive English revisions, please address this issue during revision. We propose that you use one of the editing services listed at https://www.mdpi.com/authors/english or have your manuscript checked by a colleague fluent in English writing.
Detailed response to the Editor:
The revision of Manuscript was done according to Editor’s instructions and all comments provided by Reviewers were taken into consideration and detailed improvements were done.
Reviewer #3:
In general, all parts of the article are written in understandable language and the transitions between sections are well organized. I would like to congratulate the authors for the good quality of the article, the literature reported used to write the paper, and for the clear and appropriate structure. The manuscript is well written, presented and discussed, and understandable to a specialist readership. The organization and the structure of the article are satisfactory and in agreement with the journal instructions for authors. The subject is adequate with the overall journal scope.
Reviewer #3:
Here are some suggestions I made to improve the article:
- In order to further improve the quality of the paper, an overall check of the English language is recommended.
Response:
In agreement with Reviewer #3 comment authors, language improvements were done.
- It is understood from the results that diet x muscle interactions are investigated, but analysis results are not given in the tables. It would be better if the P values of the interactions were added to the tables.
Response:
Authors appreciate the suggested by Reviewer #3, however, adding P-value for diet ×muscle interaction, when no significance was found, would not contribute to better understanding of obtained data. In addition, authors intentionally removed diet ×muscle interaction data when P > 0.05 to highlight the differences affected by diet and muscle, thus, making those effects more visible. This is related with Table 3 and Table 5. In Table 4 with color stability data, authors have presented diet ×muscle interaction data.
- Abbreviations given in the tables should be explained under the table, for example SF in Table
Response:
In agreement with Reviewer #3 comment authors made clarification of abbreviations used in Tables and Figures through the Manuscript.
- Why are the TBARS, DPPH, L, a, b and MMB values not given in the table? It should be better added to Table 3.
Response:
Authors appreciate the suggested by Reviewer #3, therefore, color data obtained during whole period of chilled-storage are presented in Table 4. However, authors kept oxidative stability (DPPH, TBARS, and WOF) figure for simplicity of presented data; these data were obtained only on Day 0 and 4 weeks.
- Figure 2, graph c has written TBARS instead of WOF, check and correct it.
Response:
In agreement with Reviewer #3 comment authors revised Figure 1C – warmed-over flavor figure.
- Only the important interactions should be shown in the figures, there is no need to give the unimportant ones. Just giving the values in the table is enough (eg DPPH).
Response:
In agreement with Reviewer #3 comment authors revised Figure 1A that presents oxidative stability data. Authors appreciate provided suggestion, however, the Figure was revised and kept.
- In the discussion, chilled storage was mentioned for color and TBARS values, but we cannot see the data related to this storage in Tables or figures. It would be nice if these results were added.
Response:
In agreement with Reviewer #3 comment, authors revised color data obtained for 3 time points (day 0, 3, and 6) and instead of Figure color data are presented in Table 4. However, authors kept oxidative stability (DPPH, TBARS, and WOF) figure for simplicity of presented data; these data were obtained only on Day 0 and after 4 weeks chilled-storage.
- The discussion is insufficient, you can improve this section with previous studies in this area.
Response:
In agreement with Reviewer #3 comment, authors have improved discussion.
Line 672-673: Similarly, Kannan et al. [19] reported that brown seaweed (Ascophyllum nodosum) extract supplemented goat diets can enhance the antioxidative status of an animal
Line 730-732: Reduced MMb content was also reported in goat meat, when an extract from brown sea-weed (Ascophyllum nodosum) was added to a goat diet [23].
Line 734-738: The antioxidant capacity of a muscle is influenced by antioxidants in seaweed, but also by the content of available minerals (Fe, Se, Zn, Cu, Mn) in the muscle [24]. Brown seaweed has shown its ability to alter antioxidant activity in ruminants [25]. However, present study indicates that S. latissima at 5% DM induced an imbalance between antioxi-dants and oxidants in lamb muscles, with a shift towards oxidative processes.
- I could not see the animal experiments ethics committee approval number in the article. It must be added.
Response:
In Line 105-107 is written: “All animal procedures for the indoor lambs were approved by the committee overseeing the rules and regulations governing animal experiments in Norway under the surveillance of the Norwegian Food Safety Authority (FOTS-ID: 16406).”
Comments on the Quality of English Language:
In order to further improve the quality of the paper, an overall check of the English language is recommended.
Response:
In agreement with Reviewer #3 comment authors, language improvements were done.
